

# Energy cost associated with moving platforms

Carolyn A. Duncan[1],*, Scott N. MacKinnon[2],*, Jacques F. Marais[3],*
and Fabien A. Basset[3]

[1] Department of Kinesiology and Integrative Physiology Michigan Tech Houghton, MI, USA
[2] Division of Maritime Studies, Chalmers University of Technology, Gothenburg, Sweden
[3] School of Human Kinetics and Recreation, Memorial University of Newfoundland, St. John's, NL, Canada
* These authors contributed equally to this work.

Corresponding author
Fabien A. Basset, fbasset@mun.ca

## ABSTRACT

**Background:** Previous research suggests motion induced fatigue contributes to significant performance degradation and is likely related to a higher incidence of accidents and injuries. However, the exact effect of continuous multidirectional platform perturbations on energy cost (EC) with experienced personnel on boats and other seafaring vessels remains unknown.

**Objective:** The objective of this experiment was to measure the metabolic ECs associated with maintaining postural stability in a motion-rich environment.

**Methods:** Twenty volunteer participants, who were free of any musculoskeletal or balance disorders, performed three tasks while immersed in a moving environment that varied motion profiles similar to those experienced by workers on a mid-size commercial fishing vessel (static platform (baseline), low and high motions (HMs)). Cardiorespiratory parameters were collected using an indirect calorimetric system that continuously measured breath-by-breath samples. Heart rate was recoded using a wireless heart monitor.

**Results:** Results indicate a systematic increase in metabolic costs associated with increased platform motions. The increases were most pronounced during the standing and lifting activities and were 50% greater during the HM condition when compared to no motion. Increased heart rates were also observed.

**Discussion:** Platform motions have a significant impact on metabolic costs that are both task and magnitude of motion dependent. Practitioners must take into consideration the influence of motion-rich environments upon the systematic accumulation of operator fatigue.

## INTRODUCTION

Motion-rich environments may have effects on the human body that can adversely impact biomechanical, physiological, and psychological aspects of vocational performance (*Crossland & Lloyd, 1993*; *Crossland, 1994*; *Wertheim, 1998*; *Duncan, MacKinnon & Albert, 2010*, *2012*). More specifically, platform motions can negatively impact upon an operator's ability to manage command, control, and communication systems, carry out

navigational tasks, perform normal ship operation and maintenance functions, and prepare food in maritime industries that involve moving platforms including but not limited to coast guard, maritime shipping and the offshore petroleum industry (*Dobie, 2003*). While the strenuous and dangerous nature of many offshore occupations is obvious, wave induced platform motions are likely responsible for accidents and injuries associated with reduced postural stability and increased work-related energy costs (EC) (*Wertheim, 1998*).

Effects on performance have been related to magnitude and frequency of the platform motions (*Colwell, 1989*; *Bles et al., 1998*; *Wertheim, 1998*; *MacKinnon et al., 2011*; *Duncan et al., 2014a*, *2014b*). Continuous exposure to motion-rich environments makes performing tasks more difficult because of the continuous postural adjustments needed to maintain balance (*Duncan et al., 2007*). These required postural adjustments will add to the EC compared to work done in a stable environment (*Baitis et al., 1995*; *Wertheim, 1998*). This increased EC may be related to motion induced fatigue (MIF). MIF has been shown to increase injury occurrences and decrease productivity in offshore workers (*Colwell, 1989*; *Haward, Lewis & Griffin, 2009*). To the authors' knowledge, current research on MIF fatigue is limited. It has been suggested MIF can be broadly characterized into two categories: fatigue due to loss or poor quality of sleep and the added EC associated with doing work in a moving environment. Unlike motion induced sickness, crew members are unable to habituate to the effects of MIF, likely resulting in accumulative effects leading to crew ineffectiveness such as lack or loss of situation awareness and errors in judgment (*Stevens & Parsons, 2002*). Accumulated fatigue can also impact upon the ability and quality of performing manual materials handling (MMH) activities in a safe and efficient manner. A better understanding of the impact of increased EC in moving environments may provide insight into the design and execution of tasks performed in maritime environments and lead to a reduction of risk of errors and injuries and improvements in safety and performance.

Accidents and injuries at sea often identify operator fatigue as the contributing cause, however, little research to date has attempted to quantify the relationship between platform motion magnitude and operator EC. *Wertheim (1998)* reported a series of earlier studies that measured EC via indirect calorimetry technique in a moving environment. All of these were done under controlled conditions within a ship motion simulator facility. A variety of activities, including standing, treadmill running, and crate stacking were examined. Results showed increases in oxygen uptake of 7% compared to no motion (NM) condition during the simulated ship movements although the increases were less than expected. Whether self-reported or observed by investigators, participants consistently reported being "severely" fatigued following testing. While a holistic approach to studying any problem is desirable, it is recognized that focused research on the mechanisms underlying MIF physiological responses is required. By addressing this knowledge gap, it is expected that such research will improve upon the understanding of the direct and indirect effects of MIF on human performance in moving environments.

The purpose of this study was to quantify the additional energy required by persons performing common tasks in simulated maritime environments as compared to stable

**Figure 1 Experimental design schematic.** Participants perform a task (i.e., quiet sitting on a chair, standing in an upright position, or lifting and lowering a load) for each of the three motion conditions. Each session (NM, LM, HM) began with quiet sitting while resting metabolic rate (RMR) was collected. Exposure to each motion condition lasted 10 min with a minimum of a 5-min rest period between conditions. The no motion trial was always presented to the participant first.

(i.e., land) environments. It is hypothesized that the continuous multi-directional perturbations will significantly increase the EC associated with performing sitting, standing and MMH tasks.

# MATERIALS AND METHODS

## Participants

Twenty (10 males, 10 females) (age: 23.4 ± 2.0 years; mass: 74.5 ± 13.1 kg; stature: 172 ± 7.0 cm; BMI: 25.1 ± 4.8 kg m$^{-2}$) participants were recruited for this study. Participants reported no susceptibility to motion sickness and were free of balance disorders and musculoskeletal diseases or injuries. Due to the influence of experience on habituation and postural control (*Duncan et al., 2016*) only individuals with no experience working in maritime environments were eligible for this study. Ethical approval for this experiment was given by the Memorial University of Newfoundland's Human Investigations Committee. All participants have signed the consent form prior to undergoing any physical tests.

## Procedures

Participants performed sitting, standing, and lifting/lowering tasks while being exposed to three motion conditions (no motion (NM), low motion (LM), high motion (HM)). The experimental protocol was completed over three sessions with at least 24 h between each session to ensure for sufficient recovery and to reduce the effects of accumulative fatigue. Each session required the participant to perform a task (i.e., quiet sitting on a chair, standing in an upright position, or lifting, and lowering a load) for each of the three motion conditions. Each session began with quiet sitting while resting metabolic rate (RMR) was collected. Exposure to each motion condition lasted 10 min with a minimum of a 5-min rest period between conditions.

While every attempt was made to randomize the task order presentation of the motion, one concession was made: the NM trial was always presented to the participant first. The NM trial was considered to be a baseline condition from which all other data collection sessions could be compared or perhaps could serve as a normalizing value. Subsequently, the LM or HM trial was randomly presented so all participants got both high and LM conditions (Fig. 1). Participants were instructed to refrain from smoking and any physical activity or consume food or drink (except water) 3 h prior to testing (*Compher et al., 2006*).

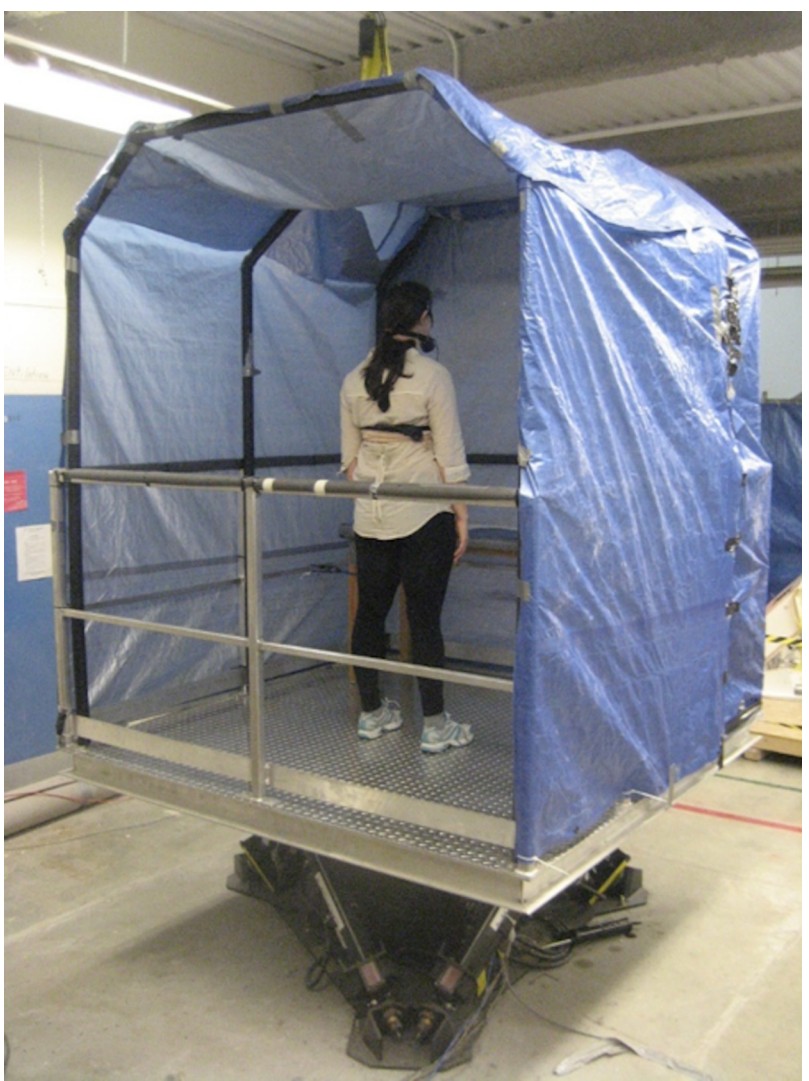

**Figure 2 Motion platform setup depicting standing.** All motion conditions were performed on a Moog 6DOF2000E electric motion platform. The platform consisted of a 2 by 2 m metal platform with 1.02 m high railings along the perimeter. A canopy enclosure eliminated external horizontal and vertical cues from the participant's field of vision. Motion conditions varied in amplitude and frequency and were derived from captured wave induced ship motions using linear wave theory that allowed for the profile to vary in magnitude (*Lloyd, 1993*). Photo credit: Carolyn A. Duncan.

## Experimental tasks

Participants required to sit on a chair (seat width: 0.48 m; seat depth: 0.48 m; seat height: 0.42 m; armrest height: 0.64 m; backrest height: 0.77 m) firmly situated to the platform with front legs in the middle of the platform facing the "bow" (front) direction for all trials. Throughout the task the participants were instructed to keep their torsos against the backrest, arms on the arm rests, and legs uncrossed with feet placed flat on the ground.

During standing trials participants remained in the middle of the motion platform in an erect standing position with arms along their sides, feet approximately shoulder width apart, weight evenly distributed between their feet and facing the bow of the platform

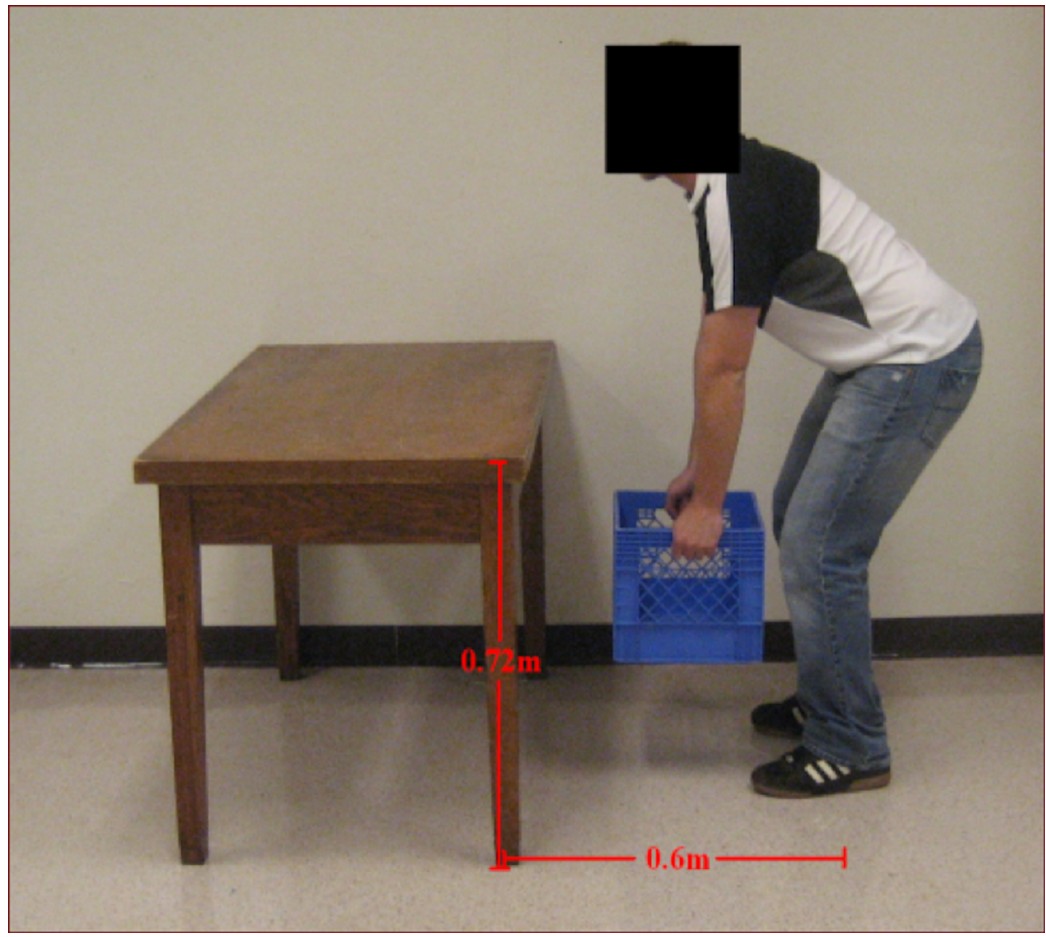

**Figure 3 Sagittal view of the lifting task that participants performed on the motion platform.** A bimanual sagittal plane lifting/lowering task required participants to lift and lower a 5.0 kg load (length: 0.327 m; width: 0.327 m, height: 0.270 m) directly to and from a table (width: 1.54 m, depth: 0.52 m; height from floor: 0.72 m) securely situated 0.60 m directly in front of the participant facing the bow direction. Photo credit: Carolyn A. Duncan.               

(Fig. 2). Stepping was permitted to correct for balance interruptions during the motion trials. Upon stepping participants were instructed to return to the original foot positions as soon as balance has been regained.

A bimanual sagittal plane lifting/lowering task required participants to lift and lower a 5.0 kg load (length: 0.327 m; width: 0.327 m, height: 0.270 m) directly to and from a table (width: 1.54 m, depth: 0.52 m; height from floor: 0.72 m) securely situated 0.60 m directly in front of the participant facing the bow direction (Fig. 3). Load size and lift dimensions were determined from the revised National Institute of Occupational Safety and Health (NIOSH) lifting equation for safe lifting (*Waters et al., 1993*). Throughout the task participants were required to keep their feet shoulder width apart and parallel to the table. Lifts and lowers were separated by 10 s intervals. The start of each lift or lower were initiated via audible signals and were performed consecutively at a rate of three lifts and lowers per minute resulting in a cumulative task rate of six manipulations per minute for a total of 60 MMH events for each condition. Participant used their own preferred lifting and lowering technique.

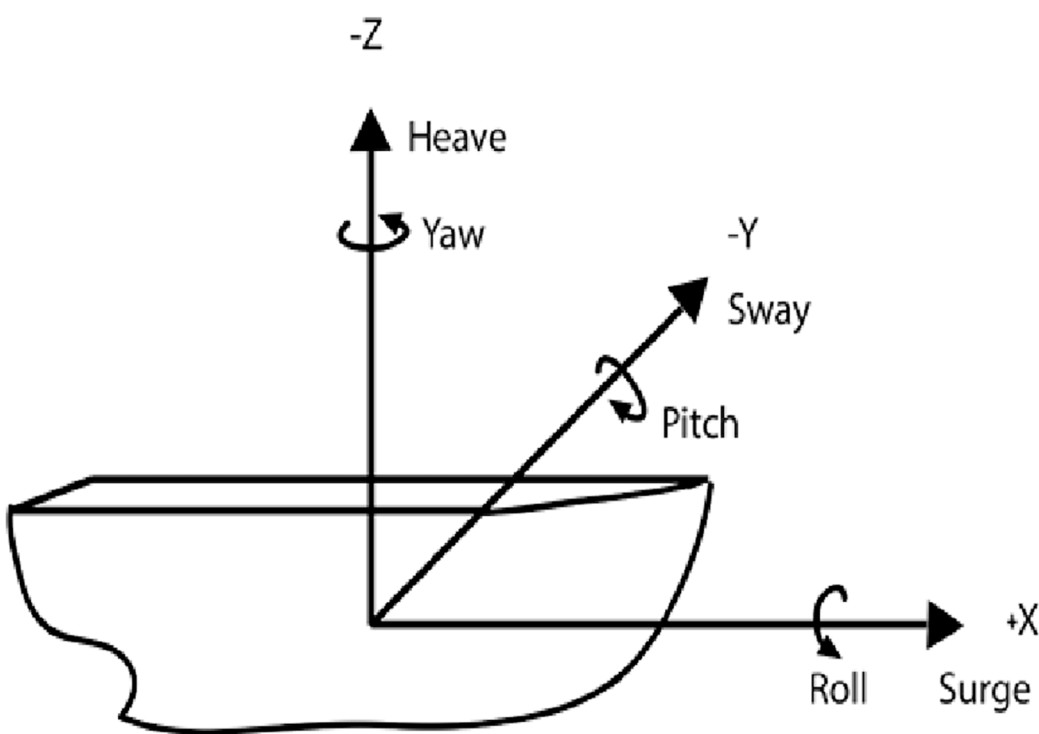

**Figure 4 Schematic of 5° of freedom ship motions.** Linear equations used to develop all motion profiles are detailed (Eqs. (1)–(5)). The low motion (LM) profile reflected a 10% increases in motion frequency relative to the original vessel from which the motion profiles were recorded and the high motion (HM) profile was characterized by a 15% increase in frequency and a 265% increase in amplitude relative to the seagoing motion profile.

## Experimental apparatus

All motion conditions were performed on a Moog 6DOF2000E electric motion platform. The platform consisted of a 2 by 2 m metal platform with 1.02 m high railings along the perimeter. A canopy enclosure eliminated external horizontal and vertical cues from the participant's field of vision. Motion conditions varied in amplitude and frequency and were derived from captured wave induced ship motions using linear wave theory that allowed for the profile to vary in magnitude (*Lloyd, 1993*). A 5° of freedom were used (roll, pitch, heave, surge, sway) (Fig. 4). Linear equations used to develop all motion profiles are detailed (Eqs. (1)–(5)). The LM profile reflected a 10% increase in motion frequency relative to the original vessel from which the motion profiles were recorded and the HM profile was characterized by a 15% increase in frequency and a 265% increase in amplitude relative to the seagoing motion profile (Table 1). Statistical analysis of differences between these conditions, performed using students' *t*-tests, confirm that the Root mean square (RMS) of the LM and HM conditions were significantly difference in pitch and roll directions ($p < 0.001$). These conditions were compared to a stable, NM condition.

$$\text{Roll}(x)(\deg/s/s) = 0.8(\sin(1.050t) + 1.25\sin(0.11t + 0.05)) \tag{1}$$

**Table 1 Motion profile characteristics including maximum, minimum and RMS of each degree of freedom for low and high motion conditions.**

| Degree of freedom | Low motion | | | High motion | | |
|---|---|---|---|---|---|---|
| | RMS | Max. | Min. | RMS | Max. | Min. |
| Sway (g) | 0.11 | 0.22 | −0.22 | 0.12 | 0.24 | −0.24 |
| Surge (g) | 0.23 | 0.44 | −0.44 | 0.25 | 0.48 | −0.48 |
| Heave (g) | 0.24 | 0.43 | 0.00 | 0.26 | 0.47 | 0.00 |
| Pitch (deg/s/s) | 3.74 | 5.30 | −5.30 | 10.24 | 14.51 | −14.50 |
| Roll (deg/s/s) | 4.42 | 6.99 | −6.99 | 11.97 | 16.97 | −16.97 |
| Yaw (deg/s/s) | 0.00 | 0.00 | 0.00 | 0.00 | 0.00 | 0.00 |

$$\text{Pitch}(y)(\deg/s/s) = 0.8\,(2.5\sin(1.76t + 0.5) + \sin(t) - 1.5) \tag{2}$$

$$\text{Heave}(G) = 0.1\,(5\sin(1.595t + 2) + 15\sin(1.21t)) \tag{3}$$

$$\text{Surge}(G) = 0.1(7.8\sin(0.649t + 4.8) + 7.8\sin(0.825t + 3.8) + 0.5) \tag{4}$$

$$\text{Sway}(G) = 0.1(18\sin(0.583t + 5) + 9\sin(1.122t + 5.4) - 0.25) \tag{5}$$

## Physiological measurements

Oxygen uptake ($\dot{V}_{O_2}$), carbon dioxide output ($\dot{V}_{CO_2}$), breathing frequency ($f_R$) and tidal volume ($V_T$) were continuously collected by an automated breath-by-breath system (Sensor Medics® version Vmax ST 1.0) using a Nafion filter tube and a turbine flow meter (opto-electric). Respiratory exchange ratio (RER) and minute ventilation ($\dot{V}_E$) were calculated as the quotient of $\dot{V}_{CO_2}$ on $\dot{V}_{O_2}$ and the product of $f_R$ and $V_T$, respectively. Heart rate values were telemetered via a Polar heart rate monitor (PolarElectro, Kempele, Finland) and recorded online. Prior to testing, gas analyzers and volume were calibrated with certified calibration gases (16.0% $O_2$ and 3.98% $CO_2$) and with a three-l calibration syringe, respectively.

## Data reduction

The RMR data were truncated by 10 min out of 20 min of data collection. This procedure discarded the first and last 5 min to nullify any metabolic rate fluctuation due to familiarization with the facemask and the expectations related to the termination of data collection. The metabolic data for sitting, standing, and lifting were truncated by 2 min out of 10 min data collection to account for the metabolic inertia and depression at the beginning and at the end data collection, respectively. The remaining segments (10 min for RMR and 8 min for the experimental conditions) were integrated and normalized over time. Oxygen uptake values were converted and expressed as EC in metabolic equivalent (MET). The same truncation and integration procedures were applied to the heart rate data. This data reduction process was undertaken to produce stable metabolic steady state information for subsequent statistical analyses. EC was computed using the following equation:

$$\text{Energy cost (MET)} = \left(\text{ml min}^{-1}\,\text{kg}^{-1}\right) \div 1\,\text{MET}\left(3.5\,\text{ml min}^{-1}\,\text{kg}^{-1}\right) \tag{6}$$

## Statistical analyses

Differences between baseline, rest periods, and tasks were analyzed using repeated measures analyses of co-variance (ANCOVA) with gender (two levels: male and female) entered as a co-variate. *Post hoc* comparisons were performed using Bonferroni corrected *t*-tests to decompose significant interaction and/or main significant effects. Assumptions of sphericity were also investigated using Mauchley's test and the adjusted Greenhouse-Geisser correction factor (epsilon ($\varepsilon$) was used to identify significance ($p \leq 0.05$) if sphericity was violated). Prior to each statistical analysis, normality and homogeneity of data sets were verified via Kolmogorov–Smirnov tests and Levene's tests, respectively. All statistical analyses were performed using Statistical Package for Social Sciences (version 21.0) (SIBM Corp., Armonk, NY, USA).

## RESULTS

### Energy cost

Results of the repeated measures ANCOVA indicated no significant differences in EC at the beginning of each trial ($p \geq 0.05$) suggesting participants started each trial with similar EC. The interaction effect between motion and gender was also not significant. Analysis of EC post-exposure revealed no significant differences in the interaction effect between motion and gender for any task or motion condition ($p \geq 0.05$). Significant differences in the main effect of motion between all conditions and all tasks (Sit: $F_{(1.88,0.35)} = 47.67$, $p < 0.001$; $d = 0.716$, 95% CI [$-0.255$ to $-0.093$]; Stand: $F_{(1.08,26.65)} = 59.16$, $p < 0.001$; $d = 0.779$, 95% CI [$-1.353$ to $-0.640$]; Lift: $F_{(1.24,18.89)} = 80.67$, $p < 0.001$; $d = 0.925$, 95% CI [$-1.155$ to $-0.509$]) (Fig. 5A). Significant differences in EC were also found between NM and LM conditions but only for the standing and lifting task.

Table 2 displays the secondary cardiorespiratory outputs ($\dot{V}_{CO_2}$, $\dot{V}_E$, $V_T$, $f_R$ and RER) as additional information; however, no statistical analyses were performed owing to the limited relevance of these parameters to the research question.

### Heart rate

Examination of the interaction effect between gender and heart rate was not statistically significant for any task ($p \geq 0.05$). Evaluation of the main effect of motion revealed statistically significant differences for standing ($F_{(1.27,759.48)} = 10.14$, $p < 0.01$; $d = 0.348$, 95% CI [$-1.728$ to $-0.806$]) and lifting ($F_{(1.35,2274.93)} = 23.10$, $p < 0.001$; $d = 0.549$, 95% CI [$-1.353$ to $-0.640$]) but not sitting ($F_{(1.65,6.10)} = 0.50$, $p = 0.58$; $d = 0.026$, 95% CI [$-0.279$ to $-0.041$]) (Fig. 5B). *Post hoc* comparisons reveal standing in LM heart rate was significantly lower than NM ($p = 0.007$; 95% CI [$-0.664$ to $-0.301$]) and HM ($p < 0.001$; 95% CI [$-2.242$ to $-0.896$]). Lifting heart rate during the HM condition was significantly greater than both NM ($p < 0.001$; 95% CI [$-2.196$ to $-1.040$]) and LM ($p < 0.001$; 95% CI [$-1.770$ to $-0.716$]).

## DISCUSSION

Work in moving environments is perceived to be much harder than performing similar tasks in stable environments. When working in moving environments, fatigue can be

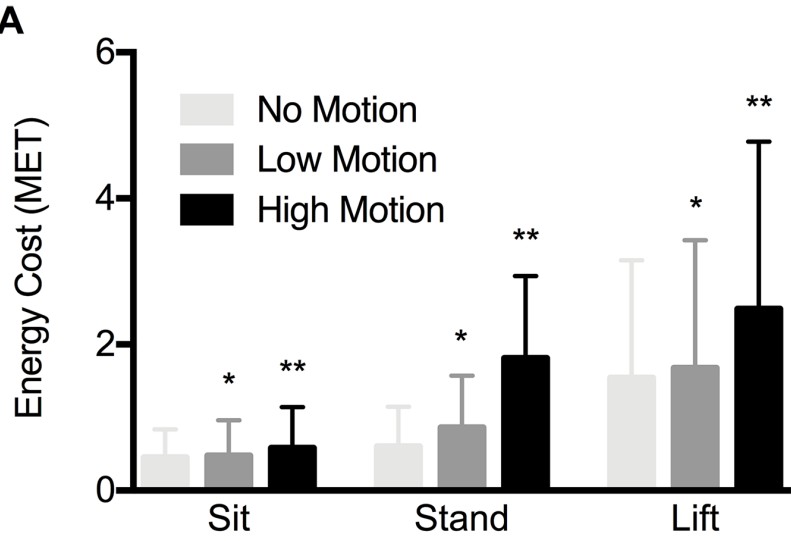

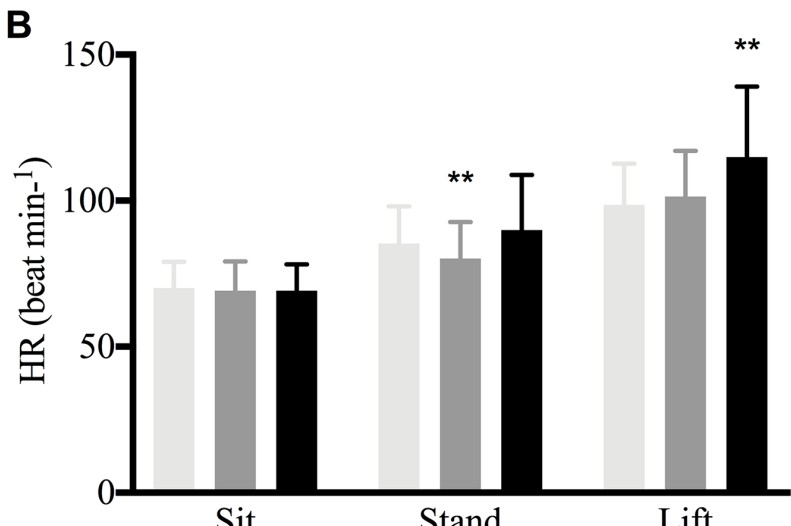

**Figure 5 Energy cost of lifting and HR responses during experimental conditions.** (A) A comparison of mean energy cost (MET) and corresponding standard deviations between motion conditions for all tasks. Statistical significant difference ($p < 0.05$) from No Motion (*) and from other two motion conditions (**). (B) A comparison of Heart rate (beat min$^{-1}$) and corresponding standard deviations between motion conditions for all tasks. Statistical significant difference ($p < 0.05$) from two motion conditions (**).                 

caused by a number of different factors including motion induced sickness (or perhaps the medications used to mediate the effects of motion sickness), motion induced loss of sleep and motion induced interruptions (MIIs)—due to the increased muscular effort needed to maintain postural stability (*Dobbins, Rowley & Campbell, 2008*). However, little research has examined the fatigue related to the EC due to motion-rich environments.

The use of simulated motion environments (*Colwell, 2005*) has provided researchers with a controlled setting in which to assess human responses to motion environments and

**Table 2  Cardiorespiratory parameters recoded during the experimental sessions with the indirect calorimetry system.**

| | SIT | | | STAND | | | LIFT | | |
|---|---|---|---|---|---|---|---|---|---|
| | NM | LM | HM | NM | LM | HM | NM | LM | HM |
| $\dot{V}_{O_2}$ (L min$^{-1}$) | 0.201 (0.05) | 0.212 (0.04) | 0.254 (0.06) | 0.256 (0.07) | 0.352 (0.11) | 0.666 (0.26) | 0.690 (0.11) | 0.757 (0.17) | 1.066 (0.29) |
| $\dot{V}_{CO_2}$ (L min$^{-1}$) | 0.180 (0.050) | 0.187 (0.042) | 0.231 (0.060) | 0.217 (0.050) | 0.299 (0.083) | 0.595 (0.244) | 0.617 (0.102) | 0.667 (0.148) | 0.993 (0.303) |
| RER (AU) | 0.89 (0.07) | 0.88 (0.07) | 0.91 (0.07) | 0.86 (0.08) | 0.86 (0.08) | 0.89 (0.09) | 0.90 (0.06) | 0.88 (0.05) | 0.93 (0.07) |
| $\dot{V}_E$ (L min$^{-1}$) | 7.1 (1.3) | 7.3 (1.4) | 7.9 (2.3) | 8.1 (1.8) | 10.5 (2.4) | 19.5 (6.6) | 18.0 (2.8) | 20.0 (4.2) | 28.9 (9.0) |

**Note:**
  Mean (SD).

allows for more controlled counterbalanced experimental designs compared to research "at sea." While research undertaken during sea trials have an element of ecological validity, a major limitation is reproducibility and control of under-foot motions between experimental conditions. Developing accurate performance prediction models require that experimental protocols avail of systematic controls of motion-defining parameters (i.e., frequency, amplitude). The simulated motions employed in this experiment were derived from sea trials and can be deemed typical of those a mariner would experience, particularly on a small near-shore vessel. It was decided to increase the frequency and/or amplitude profiles to increase the demands upon the participant, specifically to create a distinction between the LM and HM conditions as described in Table 1 (linear accelerations and angular velocities). Previous studies employed similar motion characteristics as this experiment's LM profile and reflected rather calm sea conditions (*Dobie & May, 2002*; *Heus, Wertheim & Havenith, 1998*; *Wertheim, Kemper & Heus, 2002*). The HM profile in this experiment reflects a somewhat higher sea state, with maximal linear accelerations and angular velocities at least twice the magnitude of the LM profile. Roll and pitch parameters are of particular interest since these types of motions have been shown to (independently or in combination) have the greatest effect on MIIs and EC (*Wertheim, Heus & Vrijkotte, 1994*). Maximum HM roll and pitch components were 141% and 191% larger than the LM condition's, respectively, therefore, showing that an increase in the magnitude of the sea-state will result in increased EC. These profiles are likely more reflective of larger vessel and more deep-sea locations.

Results of this study indicate that seafarers may have significantly increased EC due to working in motion-rich environments. Thus, long-term exposure to motion-rich environments will likely have a cumulative fatigue effect that can potentially lead to increased risk of falls, musculoskeletal injury, and human error. This is especially problematic in the marine context since workers can be exposed to vessel motions for extended periods of time, increasing the risk of the negative effects of motion induced fatigue (*Baitis et al., 1995*; *Colwell, 1989*). The focus of this paper was to quantify EC of persons performing three common tasks (sitting, standing and MMH) during three different motion conditions. Results indicate significant differences in EC between all motion conditions for nearly all tasks (all motion conditions were significantly greater than NM while HM was significantly greater than LM). Findings show that as motion intensity increases, so does energy demand and these outcomes fit with previous

examinations of $O_2$ uptake and EC in moving environments (*Heus, Wertheim & Havenith, 1998*; *Wertheim, Kemper & Heus, 2002*; *Breidahl et al., 2013*). However, others have reported that walking on a laterally oscillating platform can reduce EC of walking (*Joshi & Srinivasan, 2015*). These outcomes tend to show that humans may not be able to entrain to it. Nevertheless, the current article operates far from such entrainment regimes, thereby generally resulting in increased EC. In other words, there should not always be an increase in EC on a moving platform.

While statistical analyses between tasks (i.e., sitting, standing, and MMH) may be of interest, intuitively it is understood there will be differences in selected tasks' workload demands. Energy requirements will increase proportionately with increased workload (*Astrand & Rodahl, 1986*). Comparing the three tasks, energy requirements were lowest for the sitting task, followed by standing, and highest for lifting/lowering (see Fig. 3). This seems reasonable given the potential increased instability of upright bipedal stance compared with sitting in a chair with back support.

No motion sitting EC values were comparable to previous studies (*Astrand & Rodahl, 1986*; *Levine, Schleusner & Jensen, 2000*). There was a significant (25%) increase in the HM sitting EC and a significant (5%) increase in the LM sitting condition. These differences were considerably less than the standing and lifting tasks. These findings seem reasonable because during sitting the participant's centre of mass (CoM) likely remains over the base of support (BoS), predominantly defined by the area of the seat pan and foot positions on the floor (i.e., larger BoS). No correction for loss of balance are required and it is likely that the majority of any additional muscular activation relates to trunk stabilization and may primarily be related to only head/upper torso stabilization. Sitting EC may be most affected by motions involving increased forward pitch velocities since operators can rely on the backrest for high backward pitch moments. Roll motions would have a lesser effect on energy demands if arm rests are present since arm stabilization will mediate side-to-side movements. Even though relatively small sitting EC changes were observed across motions conditions, long-term exposure to motions could still induce a fatiguing effect.

No motion standing EC was also consistent with previous findings (*Garg, Chaffin & Herrin, 1978*; *Houdijk et al., 2009*; *Pandolf, Givoni & Goldman, 1977*; *Levine, Schleusner & Jensen, 2000*). Differences in NM and EC were much more apparent during standing trials, with increases of 34.7% and 157.7% for LM and HM conditions, respectively. Unlike sitting, standing creates situations for reduced postural stability and thus greater mechanical demands upon the operator. While standing in motion-rich environments workers are exposed to greater motion induced moments of inertia that directly influence an operator's ability to maintain the CoM within BoS limits. When the CoM reaches or exceeds the functional stability limits, balance is compromised, and operators have no option other than employing postural adjustments to correct for balance perturbations (*Winter, 1995*). Standing in HM states will require the operator to continually react and correct for perturbations. Previous work by *Duncan et al., (2007)* has found that as platform perturbation magnitudes increase so do the number of stepping strategies that are used to maintain balance. This increasing magnitude of postural responses would require

greater EC due to the increased muscular contractions required to move the limbs. Additionally, when not stepping to regain balance, the muscular contractions required to produce a moment to keep the CoM within the BoS to retain balance in response to larger perturbations would also be greater, and thus, requiring a higher EC.

Interestingly, the impact of motion on heart rate and EC was different. While significant changes in EC were displayed for all tasks, only the lifting task displayed a similar increasing trend with increased motion, while heart rate during LM was significantly less than both high and NMs. This is unlike previous findings (*Heus, Wertheim & Havenith, 1998*) that found the EC were accompanied by increases in heart rate; however, in their study only one motion condition was measured in both seas and simulated ocean conditions. It is plausible that during LM positive inotropic agents (e.g., catecholamines) increased ventricular contraction and, therefore, systolic ejection volume to maintain tissue oxygen perfusion at the cellular-capillary level. Future research is needed to confirm these hypotheses.

While this study examined only two motions in comparison to a baseline (NM) condition, future work should examine a broad spectrum of motions described by systematic changes in frequency and amplitude in all 6° of freedom. Such an experimental approach will allow for regression analyses to be used to describe platform motions and task-specific EC. Increased EC likely results in increased fatigue of the maritime worker, and in turn, may cause decreased ability to perform work effectively (or safely). Additionally, general fatigue may result in decreased morale of workers as the effects of motion induced fatigue are typically considered a negative subjective experience (*Shen, Barber & Shapiro, 2006*). Acute and chronic fatigue appears to be dependent on a number of factors specific to a seafaring occupation including tour length, environmental factors, extended shift length, and switching from sea to port with fatigue appearing to increase at the end of tours (*Wadsworth et al., 2006*; *Wadsworth et al., 2008*). This may lead to increased risk of injuries and errors that could lead to further accidents and injuries.

More research effort must focus on the mechanisms and outcomes of motion induced fatigue. A better understanding of motion induced fatigue will allow for improved planning of work for persons at sea for extended periods of time. While this study demonstrated increases in EC over a short period of time, this may or may not reflect extended exposure time responses. Defining the relationship between platform motions and task-specific energy demands will contribute to developing guidance notes and strategies regarding habitability, work-rest ratios and shift work scheduling, accidents and injuries, crewing and crew satisfaction, nutrition demands, and workstation and vessel design.

### Ergonomic application

Ergonomists and human factors engineers must consider these increased EC when evaluating occupational demands of seafarers. Workers in moving environments require more energy to perform equivalent tasks to those working in non-moving environments. Therefore, MMH guidelines, for example, must be sensitive to the

environment the task occurs. For example, currently the NIOSH lifting equation, assuming the baseline maximum aerobic capacity of USA workers is 9.5 kcal min$^{-1}$ (aerobic lifting capacity of an average 40-year-old female worker), workers should be lifting at no more than 50% of their maximum (4.75 kcal min$^{-1}$) for 1 h or less, 40% of their maximum (3.8 kcal min$^{-1}$) for 1–2 h; and 33% of their maximum (3.1 kcal min$^{-1}$) for 2–8 h (*Waters et al., 1993*). Conversion of the results of this study to kcal min$^{-1}$ has found that during the LM and HM condition the EC is 3.78 and 5.33 kcal min$^{-1}$, respectively. Therefore, changes to the lifting task would have to be made in order to make it safe for workers performing this same task in a moving environment. Based on the increases in EC during the standing and holding tasks, more conservative estimates would also need to be applied to all MMH tasks, including pulling pushing and carrying.

## CONCLUSIONS

This research reflects attempts to develop an experimental protocol to assess the effects of motion on the energy demands of persons working in motion-rich environments.
 From the results, it can be concluded:

1. Energy cost of common MMH tasks are greater in moving environments when compared to non-moving environments.
2. Energy cost demands are dependent upon the magnitude of the platform perturbations and the nature of the task being performed, with tasks that involve the stabilization of upright stance having higher energy demands.
3. These results provide objective outcomes on EC of commonly performed tasks by workers in maritime environments.
4. Ergonomists and human factors professionals could implement strategies based on the study outcomes to mitigate motion-induced fatigue with the aim to maintain operator performance, minimize human factors errors, and reduce work-related injuries.

## ABBREVIATIONS

**BoS**    base of support
**CoM**    centre of mass
**HM**    high motion
**LM**    low motion
**MII**    motion induced interruption
**MMH**    manual materials handling
**NM**    no motion
**RER**    respiratory exchange ratio
**RMR**    resting metabolic rate.

## ACKNOWLEDGEMENTS

The authors would like to thank the participants for their valuable cooperation to the study. We would like also to thank Dr. Thamir Alkanani for his assistance.

## Funding

This work was supported by the Natural Science and Engineering Research Council (Discovery Grant) of Canada. The funders had no role in study design, data collection and analysis, decision to publish, or preparation of the manuscript.

## Grant Disclosures

The following grant information was disclosed by the authors:
Natural Science and Engineering Research Council (Discovery Grant) of Canada.

## Competing Interests

The authors declare that they have no competing interests.

## Author Contributions

- Carolyn A. Duncan conceived and designed the experiments, performed the experiments, analyzed the data, contributed reagents/materials/analysis tools, prepared figures and/or tables, authored or reviewed drafts of the paper, approved the final draft.
- Scott N. MacKinnon conceived and designed the experiments, authored or reviewed drafts of the paper, approved the final draft.
- Jacques F. Marais performed the experiments, analyzed the data, contributed reagents/materials/analysis tools.
- Fabien A. Basset conceived and designed the experiments, analyzed the data, contributed reagents/materials/analysis tools, prepared figures and/or tables, authored or reviewed drafts of the paper, approved the final draft.

## Human Ethics

The following information was supplied relating to ethical approvals (i.e., approving body and any reference numbers):

The Human Research Ethics Authority (HREA) of Newfoundland and Labrador (Canada) granted ethical approval to carry out the experiment within its facilities.

## Data Availability

The raw data are provided in a Supplemental File.

## Supplemental Information

Supplemental information for this article can be found online at http://dx.doi.org/10.7717/peerj.5439#supplemental-information.

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
