# Peer review of "Energy cost associated with moving platforms"

_PeerJ, doi:10.7717/peerj.5439_

## Round 0.1 · original submission · Minor Revisions

Dear Authors,

Your review has now been completed for the manuscript "Energy cost associated with moving platforms". You will notice that we had three excellent and comprehensive reviews of your work and I believe that these comments can help improve the quality of this manuscript.

While the reviewers believed that the manuscript was interesting and the study relevant, you will find a general consensus that the work merits a Minor Revision. The manuscript can be further improved and streamlined in a few places. Reviewer comments can be found below. We look forward to your reply.

Regards,

·

Basic reporting

The introduction provides a good overview of the previous literature on platform motions and the impact on biomechanical, physiological, and psychological demands. Lines 67 - 76 provide a high-level descrpition of why motion-rich environments are important to examine. More detail should be provided early in the introduction as to which occupations are/would be greatly impacted. The information in these lines does not provide a clear picture to the reader which occupations the upcoming study would be most relevant to.

Line 93 - change increase to increased

Lines 103 - 105 - provide additional details as to the specific increases that were observed in oxygen uptake and what were the expected levels by the authors.

The authors provide a clear message as to the knowledge gap in this area with a clearly stated hypothesis.

Experimental design

Line 126 - should the university's entire name be listed, i.e Memorial University of Newfoundland?

Line 161 - change has to had

Line 166- add 'the' after task (throughout the task 'the' participant)

Line 166 - change 'the' to 'their'

Lines 172 - 174 - is it possible to include an image or images of the experimental setup revealing the various position the participants were placed in and the nature of the lifting tasks?

Line 183 - change increases to increase

Line 188 - suggest using significantly different throughout manuscript instead of statistically significantly difference

Line 187 - is the significant difference referring to the LM compared to the control condition and the HM compared to the control condition? Could the authors please explain why a students' t-tests were utilized instead of an ANOVA with 3 levels of conditions and 2 levels of direction?

Lines 203 – 211 – Please provide additional details as to how the VO2, VCO2, breathing frequency, and tidal volume data was analyzed. Data was collected using a breath-by-breath system, how was raw data handled, analyzed, and outliers removed?

Lines 245 – 248 – please reword sentence starting with “Significant differences in the main effect….”, confusing to follow what the significant differences are.

Validity of the findings

The authors provide a conclusion that seafarers may have significantly increased EC due to working in motion-rich environments. This conclusion is supported by the data presented on EC from each of the different conditions. However, the authors describe collecting data on VCO2, RER, breathing frequency, and tidal volume but do not represent the data in the manuscript. This information may provide additional insight into the energy requirements for the tasks during different moving environments. Please include this information or provide details as to why this data is not included.

Line 281 - provide, in parentheses, the specific tasks that were affected by the three different motion conditions.

Line 283 - end parenthesis missing after in-text citations.

Lines 335 - 356 - This paragraph is well-written but seems to be more appropriate earlier in the discussion section.

Lines 385 - 386 - please add the kcal/min for 40% and 33%, same as provided for 50%.

Line 392 - add commas to pulling pushing and carrying.

Additional comments

Thank you very much for the opportunity to review this scientific article. The authors provide a detailed description of the impacts of platform motion on the energy cost and heart rate of operators.

General Comments

Line 410 - change thanks to thank

Figure 2 - changes 10% increases to 10% increase

Figure 3 - Remove 'statistical', can state 'significant difference'; Panel B appears to be HR data and not EC data, as listed in the caption.

Reviewer 2 ·

Basic reporting

No comment. Overall clear writing. Minor suggestions for improved clarity below.

Experimental design

No comment.

Validity of the findings

The study appears sound.

Additional comments

This is a straight-forward article on the energy costs of sitting, standing, and lifting when the support surface one is standing on moves. The experiments appear to be well-designed and the results are reported thoroughly. The claims seem to follow from evidence provided.

The article is well-written overall, but I'd suggest a proof-read for minor typos. I have a few suggestions or remarks.

I suggest that there is no need to use contractions such as EC, MIF, etc. The article is short enough and these words are important enough that these contractions reduce the readability of the article. Especially EC, which is so central to the article (energy costs).

Line 177. The English alphabet X need not be used for 'times'. Please re-typeset with an appropriate symbol or perhaps this can be taken care of by the journal.

I assume that the roll-pitch-heave convention is that standard in naval architecture. As the authors know, for Euler angles, the order of the rotation matters in deciding what the current orientation of the platform is.

It was unclear to me what the exact motions applied were. The authors say in Line 182: "Linear equations used to develop all motion profiles are detailed (Equations 1-5)." It is not stated where exactly Equations 1-5 come from and how exactly their amplitudes and frequencies are modified to obtain the low motion and high motion conditions. It may be better to add variables (representing amplitude and frequencies) in Equations 1-5 and then show (as in Table 1) what the values of these amplitudes and frequencies are under different conditions. Were all 5 degrees of freedome excited simultaneously?

Perhaps as an extra panel on figure 2, I suggest that providing example plots of the applied motion for the 5 degrees of freedom.

It may be useful to mention that the motions look complex enough (e.g., pseudo-random) that humans may not be able to entrain to it. This is useful because the following article considers models of humans walking on moving platforms and suggests that under certain conditions, they may actually experience reduction in energy costs (but resulting in increased costs under other conditions).
Joshi, V. and Srinivasan, M., 2015. Walking on a moving surface: energy-optimal walking motions on a shaky bridge and a shaking treadmill can reduce energy costs below normal. In Proc. R. Soc. A (Vol. 471, No. 2174, p. 20140662).
It may be worth mentioning that the current article operates far from such entrainment regimes, thereby generally resulting in increased energy cost. In other words, it is not always the case that there should always an increase in energy costs on a moving platform.

I found the presentation of the summary data in the supplementary excel file satisfactory.

Reviewer 3 ·

Basic reporting

Clear and unambiguous, professional English used throughout.

Experimental design

Original primary research within Aims and Scope of the journal.

Validity of the findings

The results interpretation and its implications should be based on these effect sizes but not on p-values so that they provides practically meaningful conclusions.

Additional comments

Manuscript#: PeerJ-reviewing-23155-v0
Title: Energy cost associated with moving platforms

General Comments
This paper is an original work that evaluated the metabolic costs associated with maintaining postural stability in a motion-rich environment. This study was designed with scientifically sound experimental methods and approaches. However, the manuscript can be further improved and streamlined. First, the introduction is not so convincing how increase EC is directly related to fatigue-related injuries and accidents. Second, the results should provide effect sizes and its confidence intervals so that these differences are not only statistically significant, but also practically meaningful (some differences appears to be within a normal variation. Specific comments are following:

Specific comments:
Comment #1: (Introduction) As the EC and fatigue are nonlinearly related, it is less convincing how the increased EC can be directly related to fatigue or increased risks of injuries and accidents, especially in the levels found in this study.

Comment #2: (lines 163-164) Please justify how the weight and size were determined.

Comment #3: (lines 169) How many repetitions? a total of 60 lifting for 10 minute motion?

Comment #4: (lines 182-186) Please elaborate the justification for this choice. Why low motion was only justed in frequency but not amplitude while high motion was done for both.

Comment #5: (lines 186-188) Does it mean that it was practically significant? Please specify the meaningful effect sizes.

Comment #6: (line 200) Please explain the reasons. Assume it’s due to the actuator limitations?

Comment #7: (Results) Comparisons of the dependent variables (EC and HR) should include effect size and 95% CI of the effect size, not just F and p values.

Comment #7: (Conclusions) The results interpretation and conclusions should be based on these effect sizes but not on p-values.

---

## Round 0.2 · accepted · Accept

Thank you for your patience and for addressing our comments so quickly. I would agree with our reviewers that you have adequately addressed all of our concerns. I look forward to seeing your published article.

Thank you for choosing PeerJ!

·

Basic reporting

The authors have satisfied all of the review comments and made the necessary changes.

Experimental design

The authors have satisfied all of the review comments and made the necessary changes.

Validity of the findings

The authors have satisfied all of the review comments and made the necessary changes.

Additional comments

Thank you to the authors for responding to each of the comments and making the necessary changes.

Reviewer 2 ·

Basic reporting

No comment.

Experimental design

No comment.

Validity of the findings

No comment.

Additional comments

The authors have reasonably addressed all my main remarks/suggestions. This is a clearly written paper.

For some clarification questions I asked, they have clarified in their review response instead of the main paper itself, but I will leave this up to the discretion/judgement of the authors.